# Use of Ozone in Veterinary Dentistry as an Alternative to Conventional Antibiotics and Antiseptics

**DOI:** 10.3390/vetsci11040163

**Published:** 2024-04-03

**Authors:** Pierre Melanie, Carlotta Niola, Ilaria Plataroti, Simone Mancini, Filippo Fratini

**Affiliations:** 1Department of Veterinary Sciences, University of Pisa, 56124 Pisa, Italy; pierre.melanie@unipi.it (P.M.); simone.mancini@unipi.it (S.M.); filippo.fratini@unipi.it (F.F.); 2Veterinary Practiotioner, Ambulatorio Veterinario Associato “A.M.S”, 56127 Pisa, Italy; 3Veterinary Practiotioner, VetPartners, Ospedale Veterinario “Leonardo da Vinci”, 50059 Spicchio-Sovigliana, Italy; ilariaplataroti@gmail.com; 4Interdepartmental Research Center “Nutraceuticals and Food for Health”, University of Pisa, 56124 Pisa, Italy

**Keywords:** ozonated water, dogs, anti-bacterial, scaling, periodontal pathology, microbiology

## Abstract

**Simple Summary:**

Seventy-five dogs were subjected to the surgical procedures of scaling and dental extraction using conventional antibiotics and antiseptics preventive treatment or double-distilled ozonated water. Sampling by bacteriological buffer was carried out to evaluate the bacterial count in the oral cavity and evaluate the anti-bacterial efficacy of intra-operative ozone as an alternative to pre-surgical antibiotic treatment.

**Abstract:**

This paper aims to assess the disinfecting capacity of a double-distilled ozonated water solution as an alternative to common antibiotic and antiseptic devices. Seventy-five dogs were subjected to the surgical procedures of scaling and dental extraction and included in three study groups: Group 1 subjected to antibiotic pre-treatment (association amoxicillin + clavulanic acid and ampicillin + metronidazole) and disinfection with chlorhexidine, and Group 2 and 3 devoid of preventive anti-microbial treatment in which disinfection was performed, respectively, with ozonated water and chlorhexidine. Sampling by bacteriological buffer was carried out to evaluate the bacterial count in the oral cavity. The analysis of the samples determined the total mesophilic bacterial count by seeding on the culture medium via the inclusion of PCA (Plate Count Agar). The results highlighted the anti-bacterial efficacy of intra-operative ozone as an alternative to pre-surgical antibiotic treatment.

## 1. Introduction

Periodontal disease is a painful chronic inflammatory condition that involves various severity stages affecting the oral cavity and the supporting tissues of the teeth, including gums, root cementum, periodontal ligament, and alveolar bone. In dogs, periodontal disease is one of the most common dental pathologies involving complex interactions between oral microorganisms and the host immune response [1,2].

The oral microbiota consists of several aerobic, facultatively anaerobic, and strictly anaerobic bacterial species. Therefore, it is possible to state that, due to the oral cavity’s non-germ-free nature, it is impossible to attribute the responsibility of periodontal disease to a single microbial species. The site of infection is unceasingly exposed to a wide range of microorganisms and predisposing factors, such as poor oral hygiene and wet food [3,4,5].

Many studies have been presented within human medicine, identifying associations between periodontal disease and other diseases, such as diabetes mellitus, cardiovascular diseases, and immunological diseases [6].

However, it is crucial to recognize and apply the unified approach of One Health even in the development and treatment of veterinary and human oral pathologies. In this context, the phenomenon of antibiotic resistance emerges as one of the primary and growing concerns.

The rational use of anti-microbial agents for treating bacterial infections imposes a conscious choice regarding the pathogen agent. Due to the abundance of bacterial species that colonize the oral cavity daily, antibiotic use may not be as decisive as for infections in other districts. Moreover, always with a view to the rational use of antibiotic drugs, sharing the same active ingredients with humans can contribute to the increase and spread of resistance [7].

The primary categories of antibiotics utilized in veterinary dentistry encompass penicillins (ampicillin, amoxicillin and clavulanic acid), nitroimidazoles (metronidazole), and, to a lesser degree, fluoroquinolones (enrofloxacin). Despite their decreasing use, the administration of amoxicillin-clavulanic acid or metronidazole, considered for this study, is still regarded as among the most effective antibiotic options for treating oral cavity pathologies in common clinical practice as the first choice in empirical therapies [8]. Betalactams are commonly used for pre-operative prophylaxis due to their wide availability, broad spectrum of action, and ability to limit adverse effects post-operatively [9,10], while metronidazole is mostly employed in cases of refractory periodontitis [11,12].

The presence of transient bacteriemia, secondary to dental procedures in veterinary patients, is a common finding. Supporting the efficacy of amoxicillin administration, it has been shown to quickly eliminate bacteria following a routine dental extraction [13]. Documenting the predisposition of horses to develop bacteremia after tooth extraction, no complications were observed in subjects, regardless of whether they received prophylactic antibiotic treatment [14]. The same trend is detectable for dogs, where the most frequently isolated species are *Actinobacillus* spp., *Streptococcus* spp., *Actinomyces* spp., *Staphylococcus* spp., *Fusobacterium* spp., *Prevotella* spp., and *Peptostreptococcus* spp. [3,15,16]. Dental treatment may favor episodes of bacteremia, and the release of septic emboli and peri-operative antibiotic prophylaxis may be limited to cases of severe inflammation in geriatric, immuno-suppressed, or cardiopathic patients [17]. In healthy patients, however, a viable alternative is the use of other molecules locally to reduce oral bacterial load.

Chlorhexidine is the most common antiseptic agent for the disinfection of the oral cavity and skin, active especially against Gram-positive bacteria. It is applied in the salified form with gluconic acid to make it water-soluble. At low concentrations (0.02–0.06%), chlorhexidine exhibits bacteriostatic activity by altering the osmotic balance of the bacterial cell and favouring the release of potassium, phosphorus, and other low molecular weight particles. It assumes a bactericidal function at high concentrations (>0.12%), causing cell death by cytolysis [18]. As evidence of its efficacy, 0.2% chlorhexidine digluconate was shown to prevent plaque formation and the development of gingivitis in a study conducted in 1970 [19,20]. On the contrary, an in vitro evaluation conducted in 2004 examined the anti-microbial activity of chlorhexidine against endodontic pathogens (*Enterococcus faecalis*, *Staphylococcus aureus*, and *Candida albicans*) and anaerobic periodontopathogens (*Porphyromonas endodontalis*, *Porphyromonas gingivalis*, and *Prevotella intermedia*) [21].

The authors propose a new point of view to bring out some original and innovative aspects regarding the use of ozone in veterinary dental practice, specifically in the care of dog species, as a viable alternative to commonly used anti-microbials in clinical practice.

Ozone is a well-known molecule, and its effects as a disinfectant, anti-microbial, and anti-inflammatory substance are objects of various studies. It is a natural gas formed by three oxygen atoms and is a powerful oxidizing agent with a high anti-microbial power against oral pathogens [22]. Numerous studies have also reported, in fact, the therapeutic efficacy of ozone as a disinfectant, anti-microbial and anti-inflammatory substance in veterinary medicine [23,24,25].

The anti-bacterial activity of ozone is related to its capacity to react with the lipid double bonds of the bacterial membrane. Ozone promotes the peroxidation of nucleic acids and amino acids that, through the production of biologically active compounds at the cellular level, damage the membrane phospholipids and lead to the lysis of the bacterial wall. Depending on the concentration and the exposure time (>1 min), it is productive against pathogenic microorganisms, such as *P. gingivalis* and *Streptococcus mutans* [21,22]. The induction of initial chemical–physical damage of the components of the cell wall, pursued in the sensitive species and not provided with an endogenous antioxidant system, may inactivate the microorganism by the oxidation of all the essential components (enzymes, proteins, DNA, RNA) [26,27,28].

When in contact with biological fluids, ozone induces an increase of pro-oxidant species responsible for transient oxidative stress. This chemical reaction leads to the formation of aldehydes and other peroxidation products that, acting as biologically active molecules, cause, through oxidation, cell damage [29]. Radical oxygen species can induce cell apoptosis by damaging cellular components such as membrane lipids, proteins, and DNA [30]. The attribution of ozone toxicity to processes such us inhibition of intracellular enzymes, glycoproteins, glycolipids and other functional amino acids, and lipid peroxidation in the cell membrane was discussed in a 2006 study [31]. In a study comparing ozone with chlorhexidine in the treatment of *S. mutans*, the authors reported a substantial reduction of approximately 50% in initial bacterial counts in patients undergoing ozone treatment [22].

Furthermore, ozone functions as an inflammation modulator, enhancing the excretion of toxins through oxidation, stimulating the production of immuno-competent cells and immuno-globulins, improving macrophage phagocytosis, and inducing the production of various anti-inflammatory substances, such as TNF-α, leukotrienes, interleukins, and prostaglandins. These mechanisms inhibit inflammatory stimuli and promote tissue healing. Additionally, ozone enhances cellular metabolism in inflamed tissues, leading to re-epithelialization. The reduction in inflammation is attributed to the improved diffusion and transport of oxygen molecules via blood circulation [32]. Sub-gingival irrigation with ozone decreases healing time for injured tissues and reduces the count of perio-pathogenic bacteria [22].

## 2. Materials and Methods

### 2.1. Inclusion Criteria

For the present study, 75 patients belonging to the canine species were considered, without distinction of breed, age (young subjects < 5 years old, adults from 6 to 10 years old, and old > 10 years old), or sex (male/castrated 48.8%, and female/spayed 51.2%), among those conducting a visit for a dental procedure. All subjects were suffering from oral and/or periodontal infections, stomatitis, and gingivitis and had undergone scaling, with and without dental extraction.

Subjects with severe cardiac arrhythmias, uncontrolled severe hemolytic anaemias, hypoglycaemic crises, skin allergies with a vital histamine component, and high fever were not included in the study.

### 2.2. Patient Preparation and Sampling

Concerning the severity of the pathological picture, each subject was assigned an increasing score, corresponding to a mild, intermediate, or advanced stage of periodontal disease. For classification of patients, the authors considered the gingival disease index [33] and the stages of furcation involvement and exposure index [34].

Stage 1—Mild: a mild presence of tartar, halitosis, and mild gingivitis but without evidence of periodontitis, gingival retraction, gingival bleeding, increased roundness of the free gingival margin, and loss of punctuation of the gingival mucosa.

Stage 2—Moderate: a moderate presence of tartar, gingivitis with increased dulling of the free margin, mild probing bleeding, gingival retraction, mild or moderate dental mobility, and mild to moderate halitosis.

Stage 3—Severe: an abundant presence of tartar, severe hyperemia of the gum, ease of bleeding in the gum, moderate or severe gingival retraction, fragility of the alveolar bone, and increased laxity of the periodontal ligament with tooth loss.

Therefore, patients were randomly divided into three groups of 25 dogs each:-Group 1:

Subjects (*n* = 25) underwent six-days antibiotic treatment (amoxicillin + clavulanic acid 12.5 mg/kg or ampicillin + metronidazole 20 mg/kg), and peri-surgical disinfection carried out with chlorhexidine gluconate spray 2 mg/mL (Glaxosmithkline C. Healt. Corsodyl Spray).

For each patient, three samples were taken by bacteriological swab at the level of the oral cavity: pre- treatment phase (T0), post-antibiotic phase (T1), and post-treatment phase (T2).

-Group 2:

Dogs (*n* = 25) in this group did not undergo any antibiotic treatment; instead, they underwent continuous peri-surgical disinfection with ozonated double-distilled water (20 μg/mL).

This procedure reduces to only two bacteriological swabs for data collections: pre-treatment phase (T0) and post-treatment phase (T2).

-Group 3:

Dogs (*n* = 25) did not undergo antibiotic treatment before scaling, while peri-surgical disinfection was performed with chlorhexidine gluconate spray 2 mg/mL (Glaxosmithkline C. Healt. Corsodyl Spray).

As for the previous group, two samples were performed by bacteriological swab at the level of the oral cavity as: pre-treatment phase (T0) and post-treatment phase (T2).

To delineate the procedures involved in preparing the ozonated water designated for Group 2, to be utilized for continuous irrigation during scaling procedures and for the initial disinfection of the oral cavity, the process entails the preparation of ozonated double-distilled water through two cycles of continuous ozone bubbling, each lasting 10 min, at an ozone concentration of 80 μg/mL. This is aimed at achieving a minimum anti-bacterial concentration of 20 μg/mL [35]. The resulting solution is collected in a glass container and connected to the ozone generator manufactured by Herrmann Apparatebau GmbH^®^, operational at the Mario Modenato Educational Hospital in San Piero a Grado (Pisa, Italy). Although previous studies have indicated that ozone saturation in double-distilled water can be attained within five minutes [36], the bubbling duration has been extended in this study to mitigate potential concentration discrepancies.

Following preparation, the ozonated double-distilled water was stored in a refrigerator at 4 °C for approximately 24 h.

Once the surgical procedures were completed, all dogs underwent an antibiotic treatment protocol to minimize the possibility of post-operative infection.

Sampling, carried out by bacteriological swab, was performed to determine the bacterial population present in the uninfected oral cavity in the various study groups. No microscopically obvious lesions attributable to bacterial infections, such as abscesses and fistulas, were recognized in the oral cavity of the included subjects. The areas subjected to multiple sampling will affect the dental arch and calculus deposits present on it, the sublingual area, and the lateral and medial portions of the gums.

All the bacteriological swabs used for sampling were immersed in 2 mL of sterile physiological solution and stored in the refrigerator at +4 °C, waiting to be transported in a refrigerated bag to the Microbiology laboratory of the Department of Veterinary Sciences in Pisa.

### 2.3. Mesophilic Bacterial Count Determination

The collected samples were transported under refrigeration conditions as soon as possible to perform microbiological determination by inclusion in PCA (Oxoid^®^, Milan, Italy) medium to determine the total aerobic mesophilic viable bacterial count. All procedures were carried out under a laminar flow hood to exclude the chances of contamination of the samples. For each swab, 10-fold dilutions were performed in sterile saline solution. One mL of each dilution was included in PCA medium, then Petri dishes were incubated aerobically at 30 °C for 72 h.

The total mesophilic bacterial count has been expressed in log CFU/mL (Colonies Forming Units) performing to the following formula modified from ISO 7932 [37]:$$\log\;\mathrm{CFU}/\mathrm{mL}=\frac{\sum c}{V(n_{1}+0.1\;\times \;n_{2})\;d}\;+40\%$$where:-∑c represents the number of colonies recorded for each dilution.-*V* is the volume in milliliters (mL) of sterile physiological solution in which the bacteriological swab has been immersed.-$n_{1}$ denotes the number of dilutions.-d represents the dilution factor corresponding to the first dilution.

Notably, some recent scientific evidence showed that the surface of the swab collected about 60% of the material; the authors added the equivalent of 40% of the total value as a correction factor [38].

### 2.4. Statistical Analysis

The normality or non-normality of residuals was assessed using Shapiro–Wilk’s test. Student’s *t*-test was performed to assess the effect of the treatments (Groups 1, 2, and 3) on mesophilic bacterial counts considering pre-treatment phase (T0) and post-treatment phase (T2). Moreover, for Group 1 the effect of antibiotic treatment and peri-surgical disinfection were tested via Student’s *t*-tests comparing mesophilic bacterial counts of the pre-treatment phase (T0) vs. post-antibiotic phase (T1) and post-antibiotic phase (T1) vs. post-treatment phase (T2), respectively.

Statistical significance was set at 0.05. R free statistical software was used [39]. The variations between the phases T0, T1 (only for Group 1), and T2 were also calculated as the difference between the mesophilic bacterial counts expressed as logarithms of CFU/mL.

## 3. Results and Discussions

In evaluating the effectiveness of treatments, it was crucial to focus on viable bacterial load, as the number of live bacteria is what truly impacts treatment efficacy, as these bacteria are responsible for biological or clinical responses. The addition of a correction factor to account for non-viable bacteria is therefore necessary to ensure a more accurate and meaningful evaluation of results in treatment comparisons.

Based on the results obtained from the present study, it is therefore possible to justify the equivalence of ozonated water treatment alongside other microbial substances in the other study groups. This, in turn, justifies its use, considering the lesser impact on patients and fewer side effects, as discussed subsequently. Indeed, no differences were statistically reported between phases T0 and T2; moreover, for Group 1 also, no statistical differences were highlighted between T0 vs. T1 and T1 vs. T2. Therefore, the anti-microbial efficacy of all agents used was confirmed.

As shown in Figure 1, for Group 1 the difference between pre-treatment phase T0 and post-antibiotic phase T1 was 0.271 log CFU/mL, while the difference between post-antibiotic phase T1 and post-treatment phase T2 was 0.795 log CFU/mL. When the full trail was evaluated, the differences between the pre-treatment phase T0 and post-treatment phase T2 were 1.066 log CFU/mL for Group 1, 0.906 log CFU/mL for Group 2 and 0.841 log CFU/mL for Group 3.

As the oral cavity is already naturally colonized by bacteria, the aim of the antibiotic is to prevent their systemic proliferation and to avoid adverse effects on organs, such as the heart and kidneys [40,41]. Based on these considerations, the study questioned whether it was necessary to administer antibiotics prior to dental cleaning in generally healthy individuals without risk factors, who are unlikely to develop endocarditis or systemic infections.

In 25 healthy subjects who received prophylactic antibiotic treatment, a practice still in use despite its progressive decline, there was no observed effective and homogeneous reduction in microbial load. As depicted in Figure 2, after one week of treatment (phases T0–T1 of Group 1), heterogeneity in the biological response of each subject is evident. Biological response variability is to be expected considering potential differences (age, sex, race, clinical stage, etc.), so the pre-treatment with antibiotics might be not useful. This observation fully corroborates the recommendations of the guidelines that discourage the use of antibiotics before oral surgical procedures, where the risk of infection is below 5%, and severe complications are rare due to the naturally high level of bacterial contamination of the oral microbiome [16,42,43,44,45].

The feedback gained from the analysis of the data is a positive result, as it shows that the use of ozonated water allows the same or a statistically comparable result to be obtained with chlorhexidine and antibiotics.

Previous studies have confirmed the anti-microbial and disinfectant properties that demonstrate their validity with the most common anti-microbial devices [46]. Eick et al. (2012) also confirmed the bactericidal activity of ozone against *Porphyromonas gingivalis* (which, therefore, is particularly sensitive to the action of ozone), *Aggregatibacter actinomycetemcomitans*, *Fusobacterium nucleatum*, *Prevotella intermedia* and *Streptococcus sobrinus* [47].

Ahmedi et al. (2016) demonstrated ozone’s efficacy in treating dry alveolitis supported by *Streptococcus mutans* and *Actinomyces viscosus* [48]. Similarly, Sadatullah et al. (2012) confirm that ozonated water at 0.1 ppm can reduce the bacterial load of plaque formed in 24 h, reducing about 52% of anaerobic microorganisms, and 56% of streptococci [26]. The mechanism behind this reduction, according to the author, is that ozone has the dual ability to stimulate the production of ROS that inactivate bacterial cells and, through its dissociation into oxygen, to create an environment rich in O_2_ hostile to anaerobic bacterial species [26,48,49].

In human dentistry, the toxicity of ozone on oral epithelial cells and gingival fibroblasts has also been discussed. Huth et al. (2006) showed that ozonated water has less toxicity on gingival fibroblasts than chlorhexidine and hydrogen peroxide [31]. Even after dental extraction, ozone irrigation of the dental alveolus does not show adverse effects on the proliferation of periodontal ligament cells [50]. In contrast, chlorhexidine exerts its toxicity on the cell membrane, which undergoes a disruption due to non-specific electrostatic bonds with proteins and phospholipids [18,31]. Although it has an excellent anti-bacterial function, it shows the effects of flaking of the mucous membrane, slows down the re-epithelialization of lesions, and reduces the binding of fibroblasts to the tooth [51].

## 4. Conclusions

Periodontal disease in dogs represents a nexus where the health of animals, humans, and environmental sustainability intersect, underscoring the significance of adopting a One Health approach to global health management. Improper use of antibiotics in pets promotes the onset of resistance phenomena affecting the intestinal microbial flora and mucous membranes, posing a consequent risk of transmitting resistant bacteria to owners and the environment.

Antibiotic resistance is increasingly becoming a global issue among companion animals, thereby affecting their owners, highlighting the necessity of exploring alternatives to their use. In this study, the authors propose ozonized water as a viable alternative to commonly used anti-microbials in clinical practice.

The results of the observations carried out indicate that the use of ozonated water can be considered a valid alternative to current methods, as it demonstrates effects comparable to a remarkable reduction in microbial load. While antibiotics should remain the primary choice in patients at high risk of endocarditis and those with severe immuno-compromising conditions, this study’s findings suggest new and safe approaches, particularly concerning scaling and tooth extraction procedures. In such cases, prophylactic antibiotics therapy may be unnecessary and can be replaced with ozonated and chlorhexidine-based antiseptic treatments.

## Figures and Tables

**Figure 1 vetsci-11-00163-f001:**
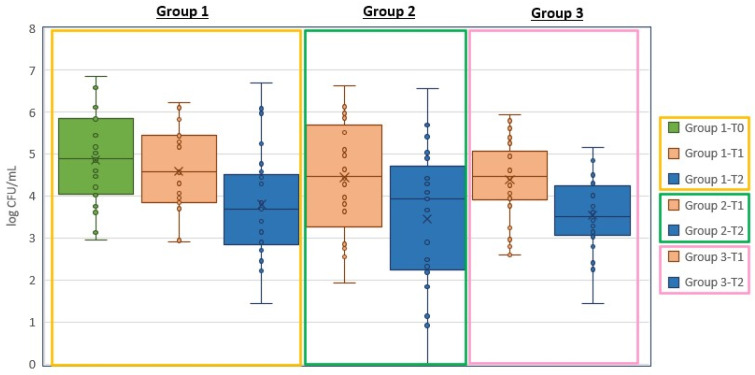
A graphic representation of the outcomes in each treatment group. Bacteriological swabs were taken at: Group 1—pre-treatment phase (T0), post-antibiotic phase (T1), and post-treatment phase (T2); Group 2 pre-treatment phase (T0) and post-treatment phase (T2); Group 3 pre-treatment phase (T0) and post-treatment phase (T2).

**Figure 2 vetsci-11-00163-f002:**
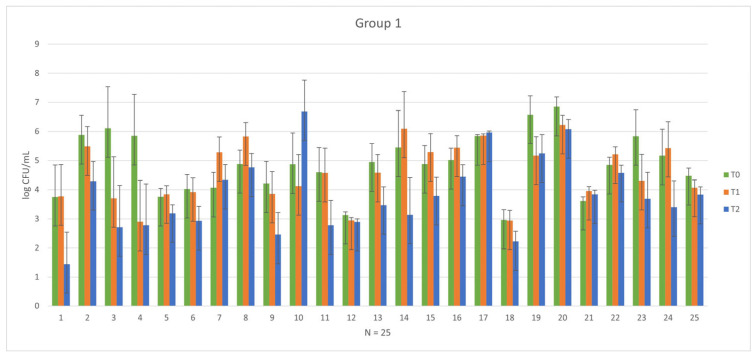
Graphical representation of the variability in pre-treatment phase (T0) and post-antibiotic phase (T1) among each subject in Group 1.

## Data Availability

All the data are available upon request from the corresponding author.

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
