# Peer review of "Use of Ozone in Veterinary Dentistry as an Alternative to Conventional Antibiotics and Antiseptics"

_vetsci, 2024, doi:10.3390/vetsci11040163_

Round 1
Reviewer 1 Report
Comments and Suggestions for Authors
The paper by Melanie et al. reports the results of evaluating the disinfectant capacity of a solution of ozonized distilled water as an alternative to common antibiotic and antiseptic compounds used during dental procedures in dogs.
The topic is interesting, and the results seem to indicate the possibility of reducing antibiotic usage. However, I have some concerns regarding the statistical analysis of the data and the presentation of results. In order for the paper to be published in the Veterinary Sciences journal, it needs modification.
For these reasons, I suggest the authors consider the following comments.
Major concerns:
L. 119-121: The age range and the male-to-female ratio should be specified.
L. 145-146: The Authors mention classifying the patients' clinical condition using a score. Is this classification arbitrary or has it been used by other Authors? Is there a reference for it?
Specify the number of dogs for each of the three stages. Is their distribution in the three experimental groups homogeneous?
L. 159-173: Specify the number of dogs in each group.
L. 160-162: Specify the duration of antibiotic treatment.
L. 182: It's unclear what the authors mean by "...uninfected oral cavity..."
L. 200: Provide a reference for the formula used. Specify the base of the logarithm. In the formula, replace "*" with "x."
L. 212-214: The Authors used parametric tests (one-way ANOVA, Student’s t-test): specify how the normality of the sample distribution was verified. It's unclear which variable was used to assess the effect of treatments with the ANOVA test and which differences were evaluated with the Student’s t-test. I also suggest moving the text from lines 220-223 to this paragraph.
L. 217-230 (Results): Overall, the results are not sufficiently clear. In particular, it's not specified what is meant by "antimicrobial efficacy" or for which variables a significant difference was not highlighted. This paragraph must be rewritten, clearly indicating what was compared (bacterial count among different groups; bacterial count within each group at different time points, etc.). It might be helpful to present the results in a table indicating the dispersion indices of the variables.
L. 252-253: The authors write "... heterogeneity in the biological response of each subject is evident, rendering it clinically irrelevant for the containment of bacterial proliferation." I believe that biological response variability is to be expected considering potential differences (age, clinical stage, etc.). However, to state that antibiotic treatment is "irrelevant for the containment of bacterial proliferation," any differences between T0 and T1 must be statistically evaluated.
Minor concerns
In many parts of the text, "ml" is written; unify it as "mL."
At the beginning of line 14, there is a "+" symbol that I don't think is necessary.
L. 71: Use italics for "in vitro."
L. 186 and 191: The transport temperature is mentioned twice.
L. 308: Delete "6. Patents."
Reviewer 2 Report
Comments and Suggestions for Authors
1、This article establishes three experimental groups: antibiotic prophylactic antibacterial plus chlorine disinfection treatment, ozone water disinfection, and chlorine disinfection treatment. I suggest that another blank control group should be established.
2、There are some issues with the logic in the introduction section. It should emphasize the prevalence and complexity of periodontal disease and its possible associations with other diseases before introducing the pathogenic and susceptibility factors of periodontal disease.
3、The introduction section mentions three main classes of antibiotics, so why was only amoxicillin selected for discussion?
4、There is a lack of literature on the effectiveness of antibiotics combined with disinfectants.
5、The introduction section is too lengthy and not direct enough. The introduction of ozonized water is overly detailed.
6、The logic in the materials and methods section is flawed. Criteria for sample selection and sample preparation can be combined.
7、It is suggested to separate the results and discussion sections.
8、There are logic issues in the conclusion section. It should first introduce why it is important to minimize the use of antibiotics and then discuss why the use of ozonized water is a better choice.
9、The years of the references are somewhat outdated. It is recommended to cite literature from recent years.
Reviewer 3 Report
Comments and Suggestions for Authors
The work is well written and the experimental design is clear. However, I recommend revising and slightly reducing the introduction: there are redundancies in a couple of places.
There are a few typos in the text that need to be corrected:
Sometimes you find "ozonized water", other times "ozonated water" please check and standardize
Line 72: the letter "t" is missing in the word Staphylococcus
Line 74: closing parenthesis missing
Round 2
Reviewer 1 Report
Comments and Suggestions for Authors
Dear Authors,
it is my opinion that the authors adequately answered the questions I had posed. The manuscript has been greatly improved.
For these reasons, I suggest accepting the manuscript in present form.
Author Response
I express my gratitude to the Reviewer for the comment and the suggestions provided, which have been necessary for improving the quality of the manuscript.